# Phosphorus Mobility in Heavily Manured and Waterlogged Soil Cultivated with Ryegrass (*Lolium multiflorum*)

**Thidarat Rupngam [1,2], Aimé J. Messiga [1,*] and Antoine Karam [2]**

[1] Agassiz Research and Development Centre, Agriculture and Agri-Food Canada, Agassiz, BC V0M 1A0, Canada; thidarat.rupngam@agr.gc.ca

[2] Department of Soils and Agri-Food Engineering, Laval University, Québec, QC G1V 0A6, Canada; antoine.karam@fsaa.ulaval.ca

\* Correspondence: aime.messiga@agr.gc.ca

**Abstract:** Extended waterlogging (WL) conditions in heavily manured soils can change soil phosphorus (P) dynamics. We assessed the effects of soil moisture regimes (field capacity (FC) and WL) and P rates on (i) dry matter (DM) yield and P offtake of ryegrass, (ii) changes in soil $Fe^{3+}$, $Fe^{2+}$, and soil P, and (iii) risk of P leaching. The treatments were tested in a four-month greenhouse experiment using intact soil columns and annual ryegrass (*Lolium multiflorum*). The DM yield and P offtake were lower under WL compared with FC. The concentration of $Fe^{3+}$ was 1984 mg kg$^{-1}$ (0–30 cm) under FC, but 1213 mg kg$^{-1}$ at 0–5 cm and 2024 mg kg$^{-1}$ at 25–30 cm depth under WL. The concentration of $Fe^{2+}$ was 244 mg kg$^{-1}$ (0–30 cm) under FC, but 2897 at 0–5 cm and 687 mg kg$^{-1}$ at 25–30 cm under WL. The water extractable P (Pw) was 12.7 mg kg$^{-1}$ at 0–5 cm and 9.5 mg kg$^{-1}$ at 25–30 cm under FC, but 8.6 mg kg$^{-1}$ at 0–5 cm and 10.5 mg kg$^{-1}$ at 25–30 cm under WL. The P saturation index (PSI) was 27.2% at 0–5 cm and 17.4% at 25–30 cm under FC, but averaged 11.9% at 0–30 cm under WL. We can conclude that extended WL associated with flooding creates reducing conditions in the soil, thus decreasing the concentration of $Fe^{3+}$, but increasing the concentrations of $Fe^{2+}$ and the solubility of P which can exacerbate the risk of P loss with runoff and leaching.

**Keywords:** reduction of $Fe^{3+}$ to $Fe^{2+}$; water extractable P; Mehlich-3 extractable P; P saturation index; P loss





## 1. Introduction

Extreme weather events associated with climate change cause heavy rainfall and subsequent flooding and waterlogging (WL) of agricultural soils [1]. In 2021, the Fraser Valley in British Columbia, Canada, was affected by an intense atmospheric river causing extreme flooding that lasted several weeks and hundreds of millions of dollars of losses [2]. In the United States, flooding was ranked second after drought among abiotic stresses contributing to monetary losses of $1.6 billion for corn (*Zea mays* L.) and soybean (*Glycine max* (L.) Merr) production in the Midwestern states in 2011 [3]. Globally, 17 million km$^2$ of land surface are affected by flooding every year [4]. Waterlogging or severe soil drainage constraints affect an estimated 10–12% of agricultural area [1]. Clearly, there is an urgent need to understand how WL affects soils and associated properties, particularly the dynamics of phosphorus (P) and metal cations such as iron, $Fe^{3+}$ and $Fe^{2+}$.

The dynamics of P can be altered under WL conditions. The availability of soil P and the concentration of dissolved P increase under WL [5]. Waterlogging increases the risk of labile P loss to surface waters. Tian et al. [6] recommended careful P fertilizer management on flooded soils to avoid excess P loss. The microbial biomass changes with increasing soil moisture content, especially the number of aerobic bacteria and subsequently the soil P [1]. In soils under WL, P mineralization rates can be altered due to anoxic conditions and changes in microbial communities. Waterlogging can also increase surface runoff and leaching as well as dissolved and particulate P losses to water bodies. The

transport of dissolved P can be enhanced under WL due to preferential flow [7]. Extended waterlogging can also affect plant morphological growth and P, potassium, nitrogen, and sugar concentration in plants' organs.

Redox reactions involving metal cations such as Fe are affected by WL. Waterlogging promotes a shift in the populations of soil microbial communities in favor of anaerobic microorganisms capable of using alternative electron acceptors such as Fe-oxyhydroxides, which are common P-fixing agents with strong chemical bonds in acidic soils. The subsequent occurrence of $Fe^{2+}$ compounds, much less effective at fixing P, enhances the solubility of P in the soil [8]. Reducing conditions associated with WL also increase soil pH by consuming soil protons and decrease the oxydo-reduction potential of the soil [5]. One consequence of the increased solubility of soil P due to changes in oxydo-reduction potential and reduction of $Fe^{3+}$ into $Fe^{2+}$ is the increased risk of P loss from soils overloaded with manure or mineral fertilizers to water bodies.

In the Fraser Valley, BC, dairy manure is a source of nutrients including P for major production systems such as forage grass and silage corn. The history of manure use and mineral fertilizer P inputs have resulted in the buildup of residual P in the soil [9]. Recent studies have shown high water extractable P in agricultural soils with concentrations above critical levels of 3.7 mg kg$^{-1}$ [10]. A study conducted in different locations in Manitoba on three anaerobic alkaline soils with varying properties showed that dissolved reactive P increases in soil pore and floodwater with manure application [11]. There is a need to understand the environmental risk associated with P release from acidic soils under WL conditions. The objective of this study was to assess the effects of soil moisture regimes (field capacity (FC) and WL) and P rates on (i) dry matter (DM) yield and P offtake of ryegrass, (ii) changes in soil $Fe^{3+}$, $Fe^{2+}$, and soil P, and (iii) risk of P leaching.

## 2. Materials and Methods

### 2.1. Site Description

The soil used for this study was sampled at the Agassiz Research and Development Centre, Agassiz, BC, Canada (49°14′34.5″ N 121°45′39.3″ W). The soil belongs to the Monroe series (Gleyed Dystric Brunisol) [12] or Typic Dystroxerepts according to U.S. Soil Taxonomy [13]. The site was covered by grass (fescues (*Festuca*), bluegrass (*Poa*), dandelion (*Taraxacum*), clover (*Trifolium*), plantain (*Plantago*), buttercups (*Ranunculus acris*), and mouse-eared chickweed (*Cerastium holosteoides*)) and was not cultivated for the past decade. The climate of the region is moderate oceanic, with relatively cool and dry summers, and warm and rainy winters. The annual rainfall varies between 1483 and 1689 mm with a peak in November at around 280–350 mm. The mean daily temperatures vary from 3.4 °C in January to 18.8 °C in August (30 years average, Agassiz CDA station).

### 2.2. Greenhouse Experiment

#### 2.2.1. Collection of Soil Columns

Twenty-four PVC pipes (40 cm height, 26.3 cm diameter, and 0.5 cm thickness) were used to collect soil columns down to 30 cm depth from the identified site. The solid PVC pipes were carefully pushed into the ground to minimize soil disturbance and preserve the original structure using a backhoe. The PVC pipes containing the soil columns were then dug out and the bottoms were lined with a fabric mesh to keep the soil undisturbed, carried to the greenhouse, and kept at room temperature. In addition, composite soil samples were collected from the different pits at 0–15 and 15–30 cm depth, sieved (<2 mm), air dried, and analyzed for general soil characteristics in a private laboratory (Table 1).

#### 2.2.2. Assembling the Lysimeters

Lysimeters were assembled using 20 L buckets (36.5 cm high and 28.2 cm diameter) to collect leachate samples from the soil columns with a valve mounted on the side wall (Figure 1). Briefly, a small PVC pipe (35 cm height and 10 cm diameter) perforated with three holes at the bottom edge was placed at the center of the bucket to support the weight

of the soil column. The three holes allowed the leachate to flow in the 20 L bucket. A 37-cm fiberglass wick (3 cm diameter) was disentangled to a length of 14 cm, spread out in a star shape and glued on a circular tray. A 4 cm diameter hole was made at the centre of the circular tray using a bi-metal hole saw through which the remaining part of the fiberglass wick was inserted. The circular tray and the fiberglass wick were placed on top of the 20 L bucket, taking care to pass the suspended part of the fiberglass wick through the small PVC pipe. The fiberglass wick acts as a hanging water column, drawing water from the undisturbed soil column without external application of suction and it has a negligible effect on contaminants [14]. A piece of landscape fabric was lined at the surface of the circular plate to act as filter for solid particles. Finally, the soil column was carefully placed on top of the circular plate and glued.

**Table 1.** General properties of the soil (0–30 cm).

|  | 0–15 cm | 15–30 cm |
|---|---|---|
| Particle size distribution (g kg$^{-1}$) |  |  |
| Sand | 585 (7) [a] | 610 (21) |
| Silt | 310 (0) | 290 (14) |
| Soil textural class | Sandy Loam | Sandy Loam |
| Soil pH | 4.2 (0.1) | 4.2 (0.0) |
| Organic matter (g kg$^{-1}$) | 43 (1.0) | 43 (1.0) |
| CEC (cmol$_{(+)}$ kg$^{-1}$) | 6.7 (3.9) | 10.2 (4.5) |
| Total carbon (g kg$^{-1}$) | 27(2.6) | 24 (1.4) |
| Total nitrogen (mg kg$^{-1}$) | 2088 (33) | 2046 (161) |
| Carbon to nitrogen ratio (%) | 13 (1.0) | 12 (1.3) |
| Total phosphorus (mg kg$^{-1}$) | 6635 (130) | 6402 (25) |
| Total organic phosphorus (mg kg$^{-1}$) | 4282 (193) | 4745 (459) |
| Total iron (g kg$^{-1}$) | 190 (1.7) | 184 (2.5) |
| Water extractable phosphorus (mg kg$^{-1}$) | 9.4 (0.9) | 9.5 (1.2) |
| Mehlich-3 phosphorus (mg kg$^{-1}$) | 214 (6) | 244 (2) |
| Mehlich-3 aluminum (mg kg$^{-1}$) | 895 (10) | 822 (19) |
| Mehlich-3 iron (mg kg$^{-1}$) | 64 (2.3) | 64 (1.0) |
| Phosphorus saturation index (%) | 18.9 (0.6) | 21.9 (0.9) |
| Oxalate phosphorus (mmol kg$^{-1}$) | 29 (0.3) | 30 (0.4) |
| Oxalate aluminum (mmol kg$^{-1}$) | 276 (1.1) | 279 (0.2) |
| Oxalate iron (mmol kg$^{-1}$) | 138 (0.6) | 137 (0.8) |
| Degree of phosphorus saturation (%) | 14 (0.1) | 14 (0.2) |

[a]: Values in parenthesis are standard deviation.

### 2.2.3. Experimental Design and Treatments

The original grass was manually cleared from each soil surface and approximately 5.0 g of Italian ryegrass (*Lolium multiflorum*) was seeded into each soil column. Standard 5TE sensors (Decagon Devices Equipment, Pullman, WA, USA) were installed in the soil columns at approximately 5 cm depth and connected to a EM50 data logger (Decagon Devices Equipment, Pullman, WA, USA) to monitor hourly volumetric water content (VWC) (Figure 2). Greenhouse temperature was maintained at $22 \pm 1\,°C$ during the day and $18 \pm 0.5\,°C$ at night and 16 h day photoperiod (05:00 a.m.–09:00 p.m.). The natural light was supplemented with 250 µmol PAR·m$^{-2}$·s$^{-1}$ artificial light (PAR, photosynthetically active radiation; Philips Ceramalux C400S51 high-pressure sodium, P.L Lighting 400 w HID) when outdoor global solar radiation was less than 300 W·m$^{-2}$. The sets of soil columns and lysimeters were arranged in a randomized complete block design including eight combinations of treatments (two soil moisture regimes and four dairy slurry P rates). The treatments were replicated thrice for a total of 24 experimental units. The two soil moisture regimes were field capacity (FC) and waterlogged (WL). The FC was set at $60 \pm 10\%$ water-filled pore space (WFPS) and the corresponding VWC was recorded using the EM50 datalogger. The FC was maintained by regular additions of tap water every time the VWC decreased by more than 10%. For the lysimeters corresponding to the WL treatments, the

4 cm diameter hole on the circular plate was sealed using a stopper to maintain water in the soil column. Soil columns were first maintained at FC for 86 days to allow germination, growth and establishment of the Italian ryegrass. Thereafter, tap water was added to soil columns corresponding to WL until a layer of water with a height of approximately 5 cm above the soil surface was obtained. The WL was then adjusted every time the height of the water decreased by approximately 2–3 cm and maintained for a total of 124 days. The four dairy slurry P rates were 0 (P0), 15 (P15), 30 (P30), and 45 (P45) kg available P ha$^{-1}$ (we assumed that 35% of dairy slurry P is available during the year of application). Dairy slurry was applied at the surface of the soil columns and allowed to infiltrate using a 1 L beaker. A supplement of N as urea (46% N) (1626, 1554, 1481, and 1409 mg N for 0, 15, 30, 45 kg available P ha$^{-1}$, respectively) was added to the soil to complement N supply with dairy slurry and meet annual forage N needs according to the local recommendation of 300 kg N ha$^{-1}$ [15]. Dairy slurry used for this experiment was collected from a manure pit at the UBC Dairy Education and Research Centre located at Agassiz RDC campus, which is applied on Agassiz RDC lands to grow forage. The dairy slurry was analyzed for total N using the Kjeldahl digestion and $NH_4$-N using distillation followed by N analysis on the FOSS Kjeltec 2400, and for total P using the microwave-assisted acid digestion adapted from EPA 3051 [16] followed by ICP analysis (Table 2).

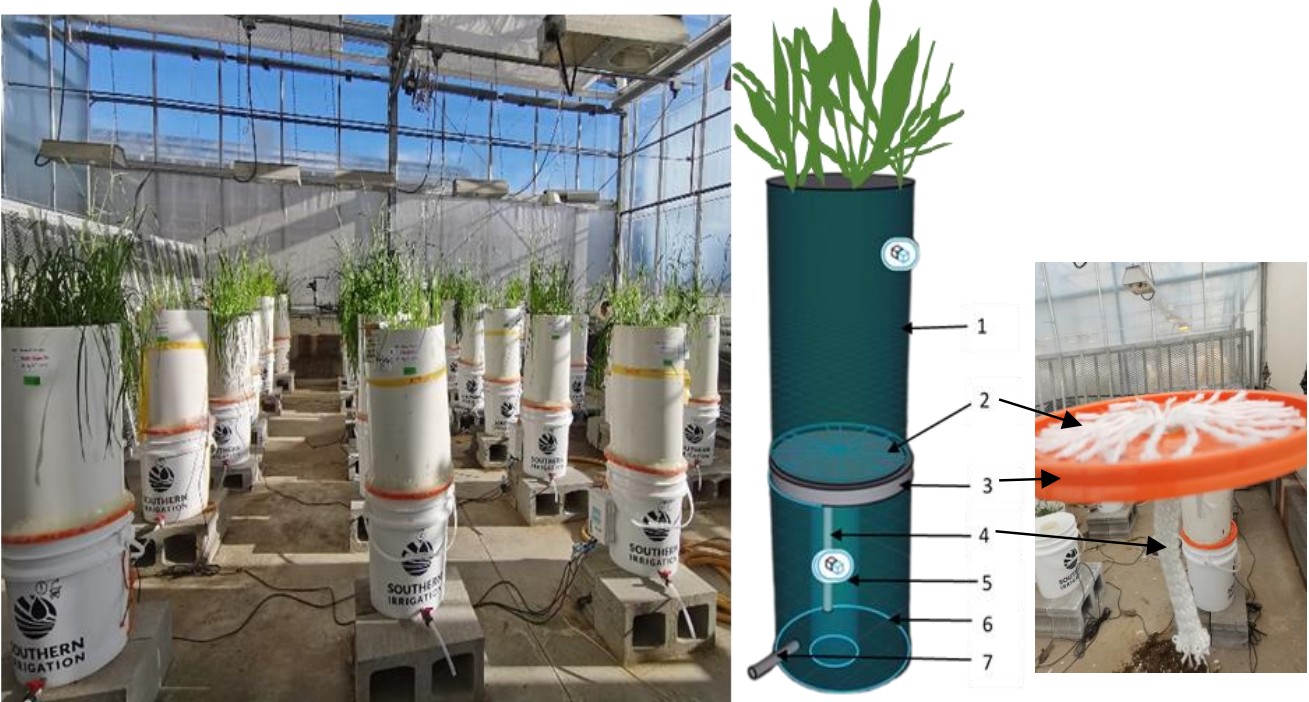

**Figure 1.** Device design used for the experiment. 1 = soil column; 2 = fiberglass wick; 3 = circular plate; 4 = suspended wick; 5 = internal column, 6 = bucket (external column), and 7 = valve.

*2.3. Plant Sampling and Analysis*

During the greenhouse experiment, plant shoot was harvested five times at maturity with the 1st, 2nd, 3rd, 4th, and 5th harvest corresponding to 42, 86, 120,162, and 210 days after seeding, respectively. Plants were cut at a height of 2 cm above the soil surface and weighed for fresh biomass. The weighed fresh biomass was oven dried at 60 °C for 72 h and weighed for dry matter (DM). The DM was ground (<1 mm) and analyzed for P. Briefly, 0.5 g of DM was digested with 10 mL of nitric acid of 68–70% (*w/w*) for 1 hr at 200 °C using a MARS 6$^{TM}$ Microwave Digestion System, Matthews, NC, USA [17]. The P concentrations were measured with an Inductively Coupled Plasma Atomic Emission Spectroscopy (ICAP 7000 series, Thermo Scientific, Cambridge, UK). The P offtake at each harvest was calculated by multiplying P concentration by DM weight.

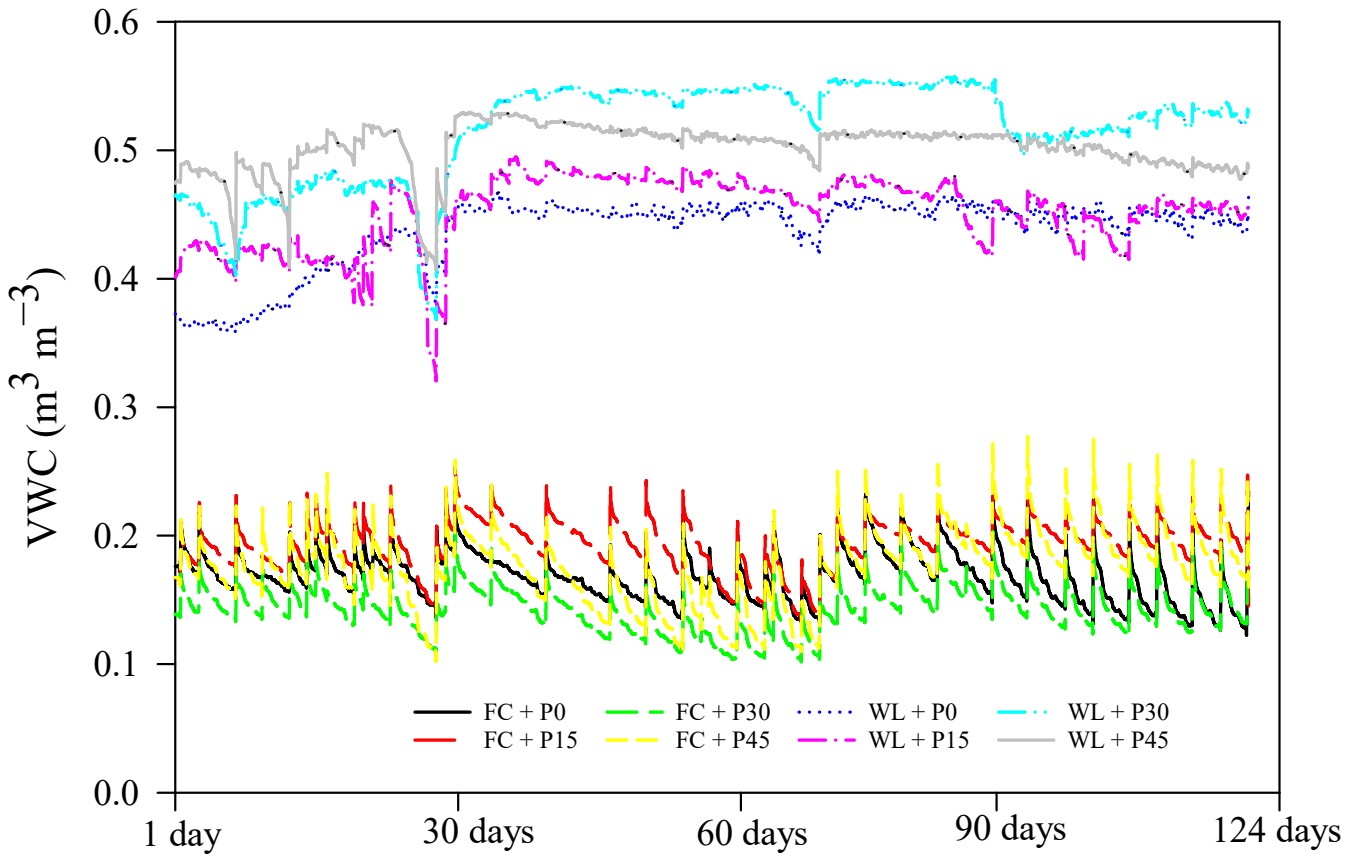

Days after submerging the soil with water

**Figure 2.** Tracking of volumetric water content (VWC) obtained by watering in soil at 0–5 cm depth after submerging the soil for treatment with waterlogging regime. FC = field capacity moisture regime; WL = waterlogging moisture regime; P0 = 0 kg P ha$^{-1}$; P15 = 15 kg P ha$^{-1}$; P30 = 30 kg P ha$^{-1}$; P45 = kg P ha$^{-1}$.

**Table 2.** Average pH, dry matter, and major nutrients in dairy slurry used in the greenhouse experiment.

|  | **Values** | |
|---|---|---|
| pH | 7.0 (0.07) [a] | |
| Dry matter (%) | 2.05 (0.18) | |
|  | Values (wet dairy slurry basis) | Values (dry dairy slurry basis) |
| Total nitrogen (mg kg$^{-1}$) | 1190.50 (0.01) | 58,897.73 (6083.51) |
| NH$_4$-N (mg kg$^{-1}$) | 598.45 (0.00) | 29,746.13 (3959.92) |
| Total phosphorus (mg kg$^{-1}$) | 206.01 (13.71) | 10,105.41 (517.21) |
| Phosphate (mg kg$^{-1}$ P as P$_2$O$_5$) | 471.76 (31.39) | 23,141.39 (1184.41) |
| Total potassium (mg kg$^{-1}$) | 890.81 (11.65) | 44,478.13 (6441.3) |

N = 5; [a]: Values in parenthesis are standard deviation.

*2.4. Soil Sampling and Analysis*

At the end of the greenhouse experiment, 10 soil cores (0–30 cm depth) were collected from every soil column using a 2 cm diameter auger. Collected soil cores were divided into six depths including 0–5, 5–10, 10–15, 15–20, 20–25, and 25–30 cm and composited per depth for a total of 144 soil samples. All soil samples were scooped in two parts, the first was air-dried and the second was kept frozen at −20 °C until subsequent analyses.

Soil pH was measured in deionized water with a 1:2 $w/v$ soil-to-solution ratio [18] using a pH meter (YSI MultiLab IDS 4010-3, Yellow Springs, OH, USA). During the course of the experiment, oxidation-reduction potential (ORP) was measured at 0–5 cm depth at 0, 10, 15, 21, 56, 99, and 124 days after submerging soil using a portable pH meter (Orion Star A221, Thermo Scientific, Ottawa, ON, Canada) and a combination of redox/ORP electrodes including (i) Orion sur-flow, (ii) ORP standards, and (iii) sure-flow reference internal fill solution (4 M KCl). In addition, the ORP for the other soil depths was estimated at the end of the experiment using the formula described by Vizier [19]:

$$ORP = 1.033 - 0.061 \times log\text{Fe}^{2+} - 0.18 \times \text{pH} \tag{1}$$

$\text{Fe}^{2+}$ is the concentration of ferrous iron (mmol $\text{L}^{-1}$), pH is the acidity of the soil depth. An underestimation of calculated ORP compared to measured ORP must be considered [5].

Total C and N were measured by dry combustion with a LECO TruMac CNS macro analyser [20]. Water extractable P (Pw) was determined by shaking 2.0 g of air-dried soil with 20 mL of deionized water for 1 h [21]. Mehlich-3 extractable P ($\text{P}_{\text{M3}}$) and other extractable cations were determined according to Ziadi and Tran [22]. Acid ammonium oxalate extractable P ($\text{P}_{\text{ox}}$), aluminum ($\text{Al}_{\text{ox}}$), and Fe ($\text{Fe}_{\text{ox}}$) were measured using the method described by Schoumans [23]. Total soil P (TP) was determined by the lithium metaborate fusion method using a Claisse M4 model fluxer (Claisse, Quebec) [24]. Total organic P (TOP) was evaluated by the ignition method [25]. The concentrations of $\text{P}_{\text{M3}}$ and other cations, TP, $\text{P}_{\text{ox}}$, $\text{Al}_{\text{ox}}$, $\text{Fe}_{\text{ox}}$ in soil extracts were measured by ICP. The concentration of Pw and TOP were quantified using the colorimetric blue method [26]. The degree of P saturation (DPS) [27] and P saturation index (PSI) [28] were calculated using the equations:

$$DPS = \frac{\text{P}_{\text{ox}}}{\alpha_m(\text{Al}_{\text{ox}} + \text{Fe}_{\text{ox}})} \times 100 \tag{2}$$

$$PSI = \frac{\text{P}_{\text{M3}}}{\text{Al}_{\text{M3}} + \text{Fe}_{\text{M3}}} \times 100 \tag{3}$$

where $\text{P}_{\text{ox}}$, $\text{Al}_{\text{ox}}$, and $\text{Fe}_{\text{ox}}$ are oxalate extractable P, Al, and Fe (mmol $\text{kg}^{-1}$); $\alpha_m$ is the maximum sorption coefficient and an average $\alpha_m$ value of 0.5 was used [29]; $\text{P}_{\text{M3}}$, $\text{Al}_{\text{M3}}$, and $\text{Fe}_{\text{M3}}$ are Mehlich 3 extractable P, Al, and Fe (mmol $\text{kg}^{-1}$).

The measurements of $\text{Fe}^{2+}$ and $\text{Fe}^{3+}$ were performed using the extraction procedure described by Yu et al. [30]. Briefly, 0.5 g of frozen soils were weighed in 15 mL centrifuge tubes containing 7.5 mL of 0.5 M HCl (1:15 $w/v$ soil to acid ratio) to prevent oxidation of $\text{Fe}^{2+}$ and shaken for 1 h on a reciprocating shaker. The suspensions were centrifuged at 4200 rpm for 10 min, decanted into a clean 15 mL centrifuge tube, and stored at 4 °C for subsequent analysis of $\text{Fe}^{2+}$ using the ferrozine method [31]. The concentration of $\text{Fe}^{3+}$ was determined following the reduction of $\text{Fe}^{3+}$ into $\text{Fe}^{2+}$ by mixing 100 µL of the supernatant, 100 µL of 10% hydroxylamine hydrochloride (100 g of hydroxylamine hydrochloride was dissolved in 1000 mL deionized water), and 2 mL of a color reagent in a 10 mL tube. The concentration of $\text{Fe}^{2+}$ was determined using a spectrophotometer at an absorbance of 562 nm using a 1 mm cuvette. The standard solutions for $\text{Fe}^{2+}$ and $\text{Fe}^{3+}$ were prepared by dissolving ferrous ammonium sulfate hexahydrate (($\text{NH}_4)_2\text{Fe}(\text{SO}_4)_2 \cdot 6\text{H}_2\text{O}$) and ferric chloride ($\text{FeCl}_3$) in 0.5 M HCl solution, respectively.

*2.5. Leachate Sampling and Analysis*

Leachates were collected at each plant harvest, and the volume was measured. A sub sample of approximately 100 mL was placed into 250 mL plastic bottles and stored at −20 °C until subsequent analyses. Leachate samples were thawed at room temperature and analyzed for inorganic P concentration using the colorimetric blue method [26]. The P leached was calculated by multiplying the concentration of P in the leachate by the volume

of the leachate collected. At the end of the experiment, leachates in the soil columns under WL were not collected which could be a limitation of this study.

### 2.6. Statistical Analyses

Statistical analyses were performed using the R programming language. Preliminary tests of normality were performed on the residuals of the model using the Shapiro-Wilk normality test. Non-normal distribution data were transformed using log, square root, exponential, and cosine functions (if needed). We analyzed the data using ANOVA with interactions followed by Tukey's multiple comparison tests (if necessary) at a significance level of 0.05 for all variables. For visualization and identification of correlated variables, the principal components analysis (PCA) was performed using the "FactoMineR" and "Factoextra" packages. In this case, individual and variable biplots were superimposed using the function "fviz_pca_biplot". The "corrplot" package was used to perform the correlation test. A significance test was performed using the function "cor.mtest", which generates *p*-values and confidence intervals for each pair of variables.

## 3. Results

### 3.1. Shoot Dry Matter Yield and P Offtake

The shoot DM yield of ryegrass decreased with the number of harvests, but the extent was influenced by the soil moisture regime ($p < 0.001$) (Figure 3a). The DM yield was 36.1 g kg$^{-1}$ during the first and 17.1 g pot$^{-1}$ during the second cuts. The DM yield during the third, fourth, and fifth cuts was 16.1, 21.9, and 18.0 g pot$^{-1}$, respectively, under FC, but 10.9, 10.7, and 12.5 g pot$^{-1}$, respectively, under WL (Figure 3a). The DM yield decreased with the number of harvests, but the extent was influenced by slurry rate ($p = 0.035$) (Figure 3b). The average DM yield increased with slurry rate ($p = 0.013$) from 17.4 g pot$^{-1}$ at P0 to 21.7 g pot$^{-1}$ at P45.

The P offtake decreased with increasing soil moisture regime, but the extent was influenced by the number of cuts ($p < 0.001$) (Figure 3c). The P offtake was 94.0 mg pot$^{-1}$ (15.16 kg P ha$^{-1}$) during the first and 38.5 mg pot$^{-1}$ (6.21 kg P ha$^{-1}$) during the second cut. The P offtake during the third, fourth, and fifth cuts was 35.2, 53.8, and 51.9 mg pot$^{-1}$ (5.68, 8.68, and 8.37 kg P ha$^{-1}$), respectively, under FC, but 23.5, 19.8, and 17.4 mg pot$^{-1}$ (3.79, 3.19, and 2.81 kg P ha$^{-1}$), respectively, under WL (Figure 3c). The cumulative P offtake increased with increasing slurry rate ($p = 0.021$). On average, it increased from 194.3 mg pot$^{-1}$ (31.34 kg ha$^{-1}$) at 0 P to 265.9 mg pot$^{-1}$ (42.89 kg ha$^{-1}$) at 45 P.

### 3.2. Change in Soil pH, ORP, and Fe Forms

Soil pH increased with increasing soil moisture regime, but the extent varied with soil depth ($p = 0.045$) (Figure 4). The soil pH under FC was 4.6 at 0–5 cm depth and averaged 4.2 at depths 5–30 cm. In contrast, the soil pH under WL was 5.1 at 0–5 cm and averaged 4.9 at depths 5–30 cm. (Figure 4). The soil pH increased with slurry application ($p < 0.001$) (Table 3). The soil pH was 4.4 at P0 and 4.6 at P15, P30, and P45.

The ORP measured at 0–5 cm soil depth decreased with increasing soil moisture regime, but the extent varied with time ($p < 0.001$) (Figure 5a). The ORP decreased from 372 at the beginning of the experiment to 266 mV under FC, but −65 mV under WL at the end of the experiment (Figure 5a). The calculated ORP at the end of experiment decreased with increasing soil moisture regime, but the extent was influenced by soil depth ($p < 0.001$) (Figure 5b). The calculated ORP was 230 mV at 0–5 cm depth and averaged 314 mV at 5–30 cm under FC, but 80 mV at 0–5 cm depth and 178 mV at 25–30 cm depth under WL (Figure 5b). The calculated ORP decreased with increasing soil moisture regime, but the extent was influenced by slurry rate ($p = 0.032$) (Figure 5c). The ORP decreased with increasing slurry rate from 322 mV at P0 and 293 mV at P45 under FC, and 185 mV at P0 to 117 mV at P45 under WL (Figure 5c).

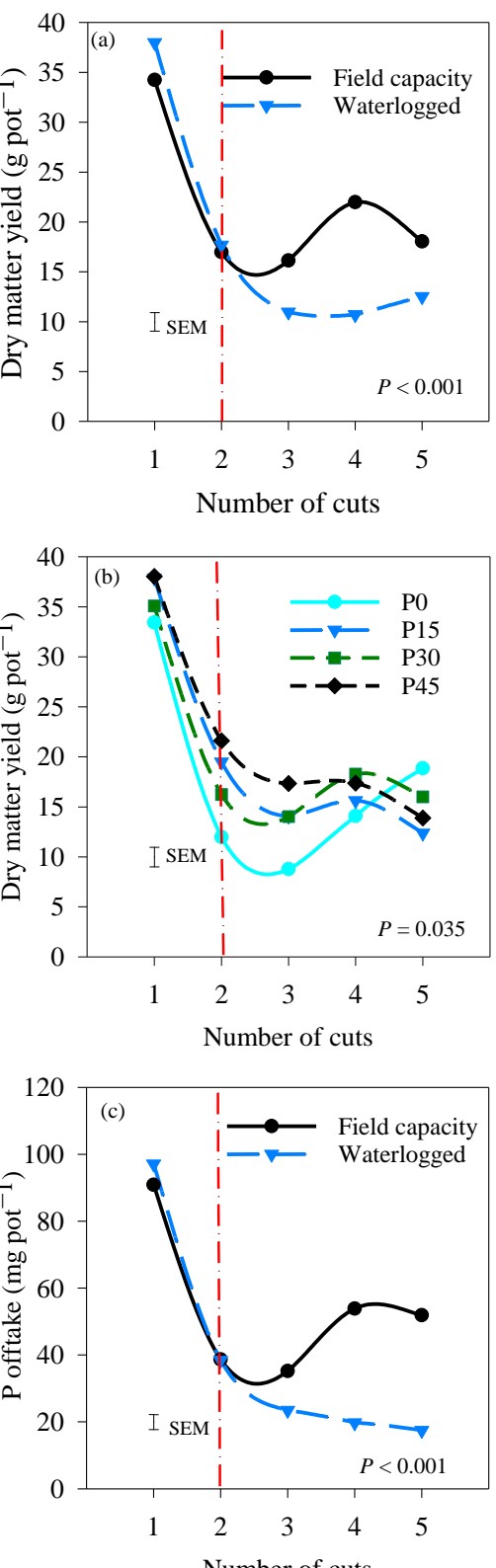

**Figure 3.** Dry matter yield of Italian ryegrass (*Lolium multiflorum*) as affected by (**a**) soil moisture regimes (field capacity and waterlogged) and (**b**) slurry rate and number of cuts. (**c**) P offtake as affected by soil moisture regimes and number of cuts. P0 = 0 kg P ha$^{-1}$; P15 = 15 kg P ha$^{-1}$; P30 = 30 kg P ha$^{-1}$; P45 = kg P ha$^{-1}$. Error bars represent standard errors of the means (SEM; *n* = 120; df = 80). Red dashed line represent the period where soil was submerged at 2nd cut.

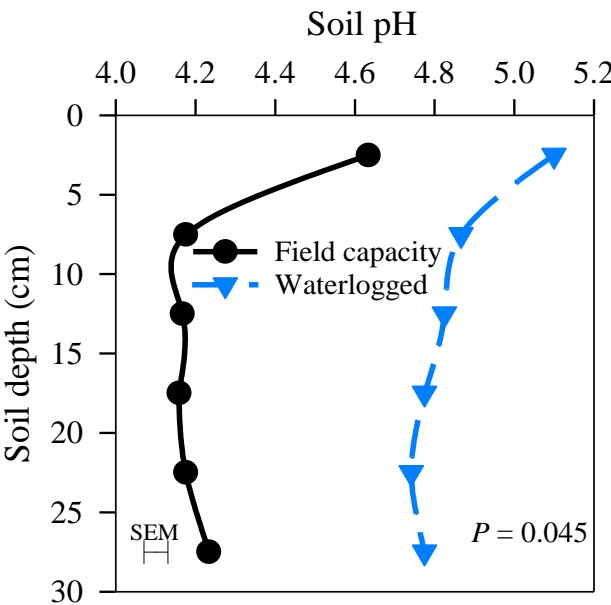

**Figure 4.** Effects of soil moisture regime (field capacity and waterlogged) and soil depth (0–30 cm) on soil pH after 124 days of submerging. Error bars represent standard errors of the means (SEM; *n* = 144; df = 96).

**Table 3.** Soil properties as affected by moisture content, slurry rate, and soil depth after 124 days of submerging soil under annual ryegrass cultivation.

| | Soil pH | Calculated ORP | $Fe^{3+}$ | $Fe^{2+}$ | Pw | $P_{M3}$ | PSI | TC | TN | C-to-N | TP | TOP | $Fe_{M3}$ |
|---|---|---|---|---|---|---|---|---|---|---|---|---|---|
| | | | | | Moisture content (MC) | | | | | | | | |
| Field capacity | 4.3 b | 297.9 a | 1990.3 a | 242.2 b | 10.3 a | 181.8 a | 20.2 a | 25,686.2 a | 2182.6 a | 12.1 a | 6385.6 a | 4887.7 a | 69 b |
| Waterlogged | 4.8 a | 139.9 b | 1535.4 b | 2356.6 a | 9.3 a | 97.6 b | 11.9 b | 25,386.4 a | 2254.3 a | 11.5 a | 6334.0 a | 4709.3 b | 196 a |
| SEM (MC) | 0.02 | 3.91 | 39.15 | 74.70 | 0.34 | 4.96 | 0.31 | 17.7 | 4.22 | 0.98 | 7.89 | 96.09 | 0.25 |
| | | | | | Slurry (S) | | | | | | | | |
| 0 | 4.4 b | 253.8 a | 1862.2 a | 890.9 b | 7.9 b | 159.4 a | 17.2 a | 24,941.1 b | 2117.4 b | 11.9 a | 6410.9 a | 4594.3 a | 124 b |
| 15 | 4.6 a | 212.1 b | 1732.9 a | 1272.7 ab | 10.4 ab | 150.7 a | 15.7 b | 25,988.2 a | 2403.8 a | 11.2 a | 6393.9 a | 4873.4 a | 130 ab |
| 30 | 4.6 a | 204.6 b | 1776.2 a | 1459.2 ab | 11.3 a | 142.1 a | 16.8 ab | 25,813.1 a | 2162.1 ab | 12.2 a | 6364.9 a | 4647.7 a | 135 a |
| 45 | 4.6 a | 205.2 b | 1680.3 a | 1574.8 a | 9.5 ab | 106.8 b | 14.6 bc | 25,402.8 ab | 2190.5 ab | 11.8 a | 6269.5 a | 5075.4 a | 141 a |
| SEM (S) | 0.03 | 7.56 | 45.31 | 134.67 | 0.33 | 5.85 | 0.46 | 38.9 | 10.59 | 0.98 | 7.99 | 95.54 | 0.62 |
| | | | | | Soil depth (SD) | | | | | | | | |
| 0–5 cm | 4.9 a | 154.9 c | 1588.9 b | 1573.5 ab | 11.5 a | 145.7 a | 19.7 a | 32,846.8 a | 2805.9 a | 11.9 a | 6629.8 a | 5210.5 a | 144 a |
| 5–10 cm | 4.5 b | 214.7 b | 1663.8 ab | 2075.9 a | 7.9 a | 123.5 a | 15.1 b | 27,348.3 b | 2395.7 b | 11.6 a | 6502.8 a | 5056.8 ab | 160 a |
| 10–15 cm | 4.5 b | 221.4 ab | 1528.1 b | 1848.1 a | 8.9 a | 134.7 a | 15.0 b | 24,589.6 c | 2193.4 bc | 11.5 a | 6415.5 a | 4813.4 ab | 145 ab |
| 15–20 cm | 4.5 b | 238.9 a | 1861.3 ab | 946.1 bc | 9.9 a | 141.9 a | 15.5 b | 23,308.2 d | 2076.0 c | 11.6 a | 6064.7 a | 4721.7 ab | 128 b |
| 20–25 cm | 4.5 b | 243.2 a | 1891.0 ab | 889.7 bc | 10.3 a | 144.1 a | 15.5 b | 22,682.3 d | 1921.4 c | 12.2 a | 6306.8 a | 4537.4 ab | 113 c |
| 25–30 cm | 4.5 b | 240.3 a | 2044.2 a | 463.1 c | 10.0 a | 148.5 a | 15.8 b | 22,442.6 d | 1918.2 c | 12.0 a | 6239.3 a | 4451.4 b | 105 c |
| SEM (SD) | 0.03 | 7.32 | 42.19 | 118.65 | 0.33 | 6.12 | 0.44 | 333.96 | 28.28 | 0.16 | 7.74 | 74.75 | 0.59 |
| | | | | | *p*-values | | | | | | | | |
| MC | <0.001 | <0.001 | <0.001 | <0.001 | ns | <0.001 | <0.001 | ns | ns | ns | ns | 0.039 | <0.001 |
| S | <0.001 | <0.001 | ns | 0.024 | 0.005 | ns | <0.001 | 0.003 | 0.021 | ns | ns | ns | 0.004 |
| MC × S | ns | 0.032 | ns | 0.023 | 0.003 | 0.264 | <0.001 | ns | 0.039 | ns | ns | ns | ns |
| SD | <0.001 | <0.001 | 0.002 | <0.001 | ns | <0.001 | <0.001 | <0.001 | <0.001 | ns | ns | 0.005 | <0.001 |
| MC × SD | 0.045 | <0.001 | 0.027 | <0.001 | 0.054 | <0.001 | <0.001 | <0.001 | ns | ns | ns | ns | 0.002 |
| S × SD | ns | ns | ns | ns | ns | ns | ns | ns | ns | ns | ns | ns | ns |
| MC × S × SD | ns | ns | ns | ns | ns | ns | ns | ns | ns | ns | ns | ns | ns |
| N | 144 | 144 | 144 | 144 | 144 | 144 | 144 | 144 | 144 | 144 | 144 | 143 | 144 |

ns, not significant at *p* value > 0.05; SEM, standard errors of the means; Calculated ORP (mV); $Fe^{3+}$, concentration of $Fe^{3+}$ (mg kg$^{-1}$); $Fe^{2+}$, concentration of $Fe^{2+}$ (mg kg$^{-1}$); Pw, water extractable P (mg kg$^{-1}$); $P_{M3}$, Mehlich-3 extractable P (mg kg$^{-1}$); PSI, phosphorus saturation index (%); TC, total carbon (mg kg$^{-1}$); TN, total nitrogen (mg kg–1); C-to-N, carbon to nitrogen ratio (%); TP, total P (mg kg$^{-1}$); TOP, total organic P (mg kg$^{-1}$); $Fe_{M3}$, Mehlich-3 extractable iron (mg kg$^{-1}$).

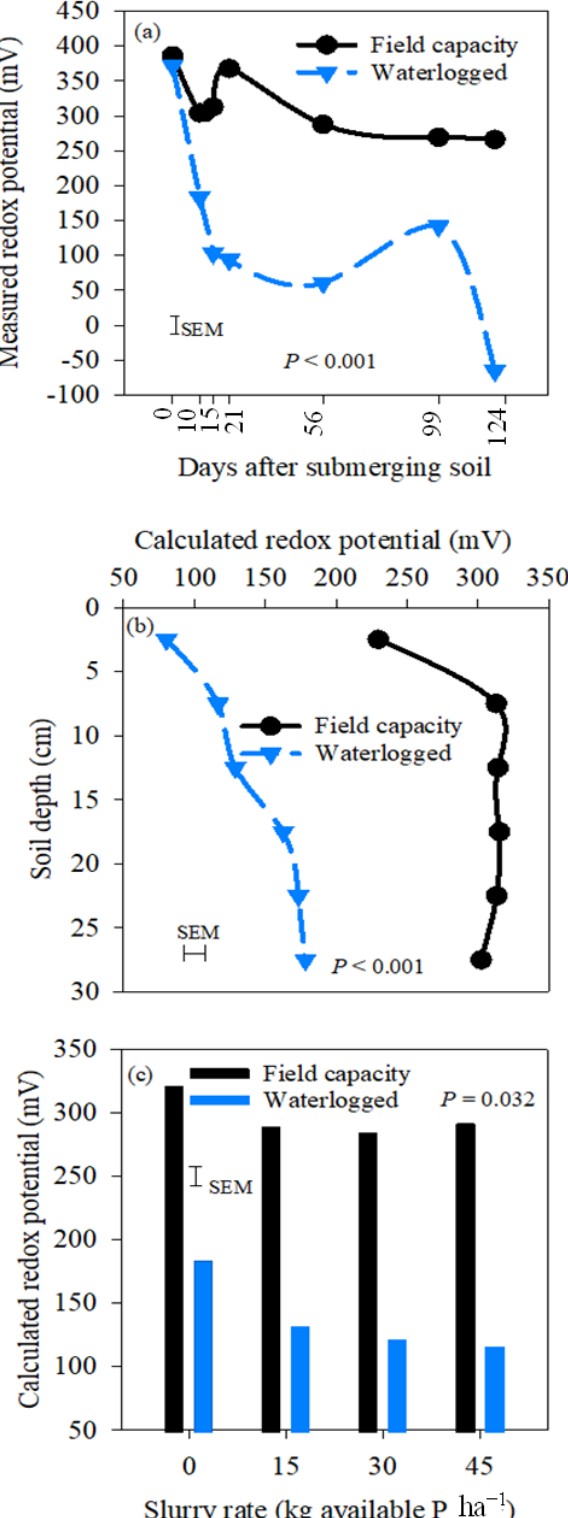

**Figure 5.** (**a**) Evolution of redox potential measured at 0–5 cm soil depth using redox/ORP electrode as affected by soil moisture regimes (field capacity and waterlogged) throughout 124 days. Error bars represent standard errors of the means (SEM; *n* = 167; df = 111). Effects of soil moisture regime and soil depth (0–30 cm) on (**b**) calculated redox potential and slurry rate on (**c**) calculated redox potential after 124 days of submerging. Error bars represent standard errors of the means (SEM; $n_{b,c}$ = 144; $df_{b,c}$ = 96).

The concentration of $Fe^{3+}$ decreased with increasing soil moisture regime, but the extent was influenced by soil depth ($p = 0.027$) (Figure 6a). The concentration of $Fe^{3+}$ varied between 1904 and 2064 mg $kg^{-1}$ from 0 to 30 cm depth under FC, but increased from 1213 at 0–5 cm to 2024 mg $kg^{-1}$ at 25–30 cm soil depth under WL (Figure 6a). Slurry rate did not affect the concentration of $Fe^{3+}$ (Figure 6b). The concentration of $Fe^{2+}$ increased with increasing soil moisture regime, but the extent was influenced by soil depth ($p < 0.001$) (Figure 6c) and slurry rate ($p = 0.023$) (Figure 6d). The concentration of $Fe^{2+}$ was on average 244 mg $kg^{-1}$ at 0–30 cm soil depth under FC, but decreased from 2897 mg $kg^{-1}$ at 0–5 cm depth to 687 mg $kg^{-1}$ at 25–30 cm depth under WL (Figure 6c). The concentration of $Fe^{2+}$ increased from 1535 mg $kg^{-1}$ at P0 to 2908 mg $kg^{-1}$ at P45 under WL (Figure 6d).

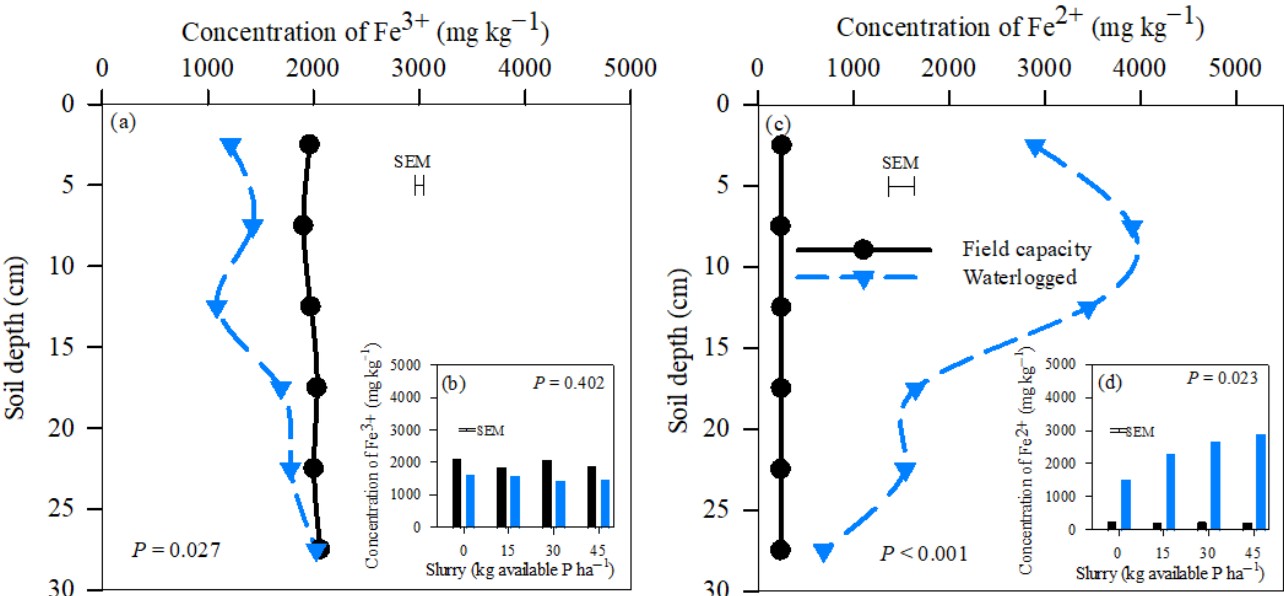

**Figure 6.** Effects of soil moisture regimes (field capacity and waterlogged) and soil depth on the concentration of (**a**) ferric ($Fe^{3+}$) and (**c**) ferrous ($Fe^{2+}$) after 124 days of submerging. Effects of soil moisture regime and slurry rate on the concentration of (**b**) ferric ($Fe^{3+}$) and (**d**) ferrous ($Fe^{2+}$) after 124 days of submerging. Error bars represent standard errors of the means (SEM; *n* = 144; df = 96).

*3.3. Changes in Pw, $P_{M3}$, PSI, and TOP within the Soil Depth Profile*

The concentration of Pw decreased with increasing soil moisture regime, but the extent was influenced by soil depth ($p = 0.054$) (Figure 7a) and slurry rate ($p = 0.003$) (Figure 7b). The Pw was 12.7 mg $kg^{-1}$ at 0–5 cm and 9.5 mg $kg^{-1}$ at 25–30 cm soil depth under FC, but 8.6 mg $kg^{-1}$ at 0–5 cm and 10.5 mg $kg^{-1}$ at 25–30 cm soil depth under WL (Figure 7a). The Pw averaged 9.5 mg $kg^{-1}$ across slurry rates under WL, but increased with increasing slurry rates from 6.8 mg $kg^{-1}$ at P0 to 13.7 mg $kg^{-1}$ at P30 under FC (Figure 7b).

The concentration of $P_{M3}$ decreased with increasing soil moisture regime, but the extent was influenced by soil depth ($p = 0.003$) (Figure 7c). The $P_{M3}$ decreased from 210 mg $kg^{-1}$ at 0–5 cm to 162 mg $kg^{-1}$ at 25–30 cm under FC, but increased from 81 mg $kg^{-1}$ at 0–5 cm to 137 mg $kg^{-1}$ at 25–30 cm soil depth under WL. The $P_{M3}$ decreased with increasing slurry rate (Table 3). The $P_{M3}$ decreased from 159.4 mg $kg^{-1}$ at P0 to 106.8 mg $kg^{-1}$ at P45.

The PSI decreased with increasing soil moisture regime, but the extent was influenced by soil depth ($p < 0.001$) (Figure 7e). The PSI was 27.2% at 0–5 cm and 17.4% at 25–30 cm soil depth under FC, but averaged 11.9% at 0–30 cm soil depth under WL (Figure 7e). The PSI decreased with increasing soil moisture regime, but the extent was influenced by slurry rate ($p < 0.001$) (Figure 7f). The PSI averaged 21% across the slurry rates under FC, but decreased from 14.4% at P0 to 9.7% at P45 under WL (Figure 7f).

The TOP decreased with increasing soil moisture regime ($p = 0.039$) (Table 3). The TOP was 4887.7 mg $kg^{-1}$ under FC but 4709.3 mg $kg^{-1}$ under WL. The TOP decreased with

increasing soil depth ($p$ = 0.005) (Table 3). The TOP was 5210.5 mg kg$^{-1}$ at 0–5 cm and 4451.4 mg kg$^{-1}$ at 25–30 cm.

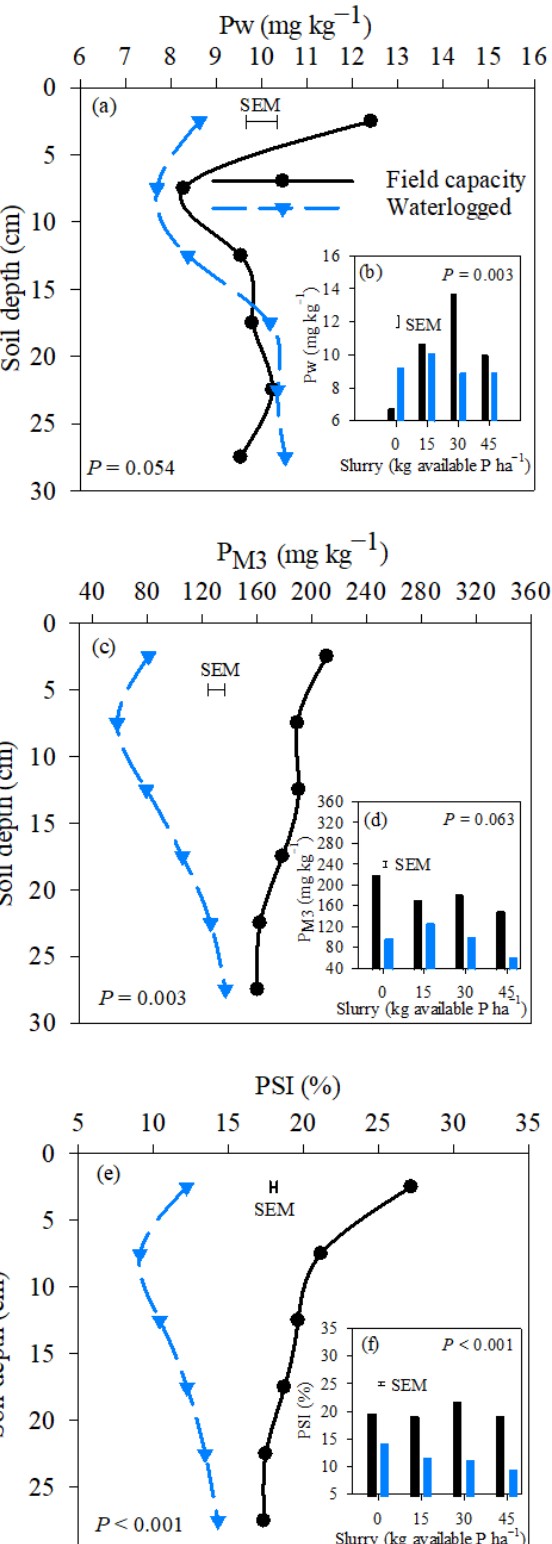

**Figure 7.** Effects of soil moisture regimes (field capacity and waterlogged) and soil depth on (**a**) water extractable phosphorus (Pw), (**c**) Mehlich-3 extractable phosphorus (P$_{M3}$), and (**e**) phosphorus saturation index (PSI). Effects of soil moisture regimes and slurry rate on (**b**) Pw, (**d**) P$_{M3}$, and (**f**) PSI. Error bars represent standard errors of the means (SEM; n$_{a,b,c,d,e,f}$ = 144; df$_{a,b,c,d,e,f}$ = 96).

### 3.4. P Loss in Leachate under Field Capacity Regime

Analysis of Variance (ANOVA) did not show any significant difference in the concentration of P loss in leachate between the slurry rate and time of collection (Figure 8). The concentration of P loss in leachate averaged 0 μg (0 mg L$^{-1}$), 258.1 μg (0.019 mg L$^{-1}$), 211.2 μg (0.039 mg L$^{-1}$), and 743.2 μg (0.131 mg L$^{-1}$) at 0 P, 15 P, 30 P, and 45 P, respectively.

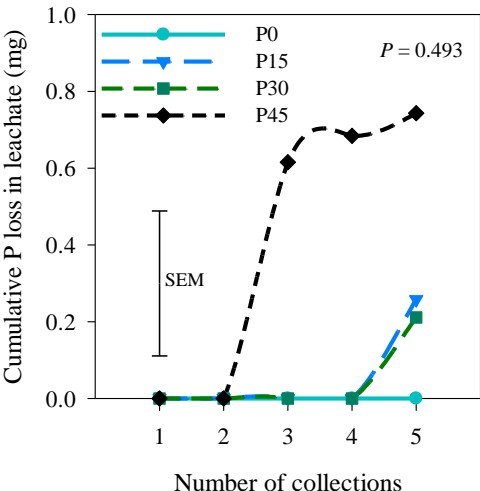

**Figure 8.** Cumulative available P loss in leachate under field capacity regime as affected by dairy slurry rates and number of collections. P0 = 0 kg P ha$^{-1}$; P15 = 15 kg P ha$^{-1}$; P30 = 30 kg P ha$^{-1}$; P45 = kg P ha$^{-1}$. Error bars represent standard errors of the means (SEM; *n* = 60; df = 40).

### 3.5. Relationship between the Soil Parameters

Principal component analysis demonstrated that the studied soil parameters were divided into two groups: (i) the variables that were related to the field capacity regime and (ii) the variables that were related to the waterlogged moisture regime (Figure 9). The ORP, Fe$^{3+}$ concentration, P$_{M3}$, PSI, and DPS were related to the field capacity moisture regime. The soil pH values and Fe$^{2+}$ concentrations were related to the waterlogged moisture regime.

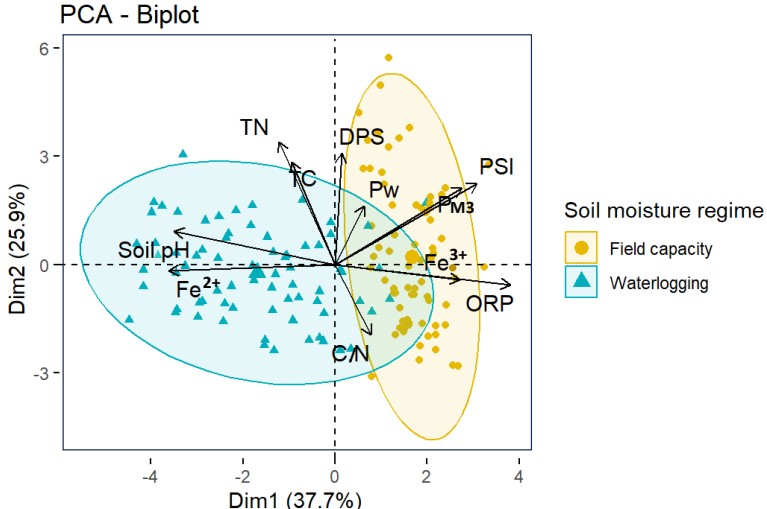

**Figure 9.** Principal component analysis (PCA)—Biplot reveals the projection of the variables, grouped according to soil moisture regime (field capacity and waterlogged). ORP = oxidation−reduction potential; Fe$^{3+}$ = ferric concentration; Fe$^{2+}$ = ferrous concentration; P$_{M3}$ = Mehlich−3 extractable phosphorus; Pw = water extractable phosphorus; PSI = phosphorus saturation index; DPS = degree of phosphorus saturation; TC, TN = total C and N; Soil.pH = soil pH; and C/N = C/N ratio.

## 4. Discussion

### 4.1. Effects of Soil Moisture Regimes, Slurry Rate, and Duration of Plant Growth on Shoot Dry Matter Yield and P Offtake

The decrease in shoot DM yield of ryegrass under the waterlogged moisture regime can be attributed to physiological and metabolic changes of ryegrass. The limited concentration of oxygen under WL conditions [1] triggers the fermentation process for ATP production, which is inefficient compared with the respiration process [32]. Soil waterlogging stress can have a significant negative impact on root growth and functions and sometimes leads to root decay, which limits the volume of the rhizosphere of plants. The formation and proliferation of aerenchyma in adventitious roots observed on ryegrass under WL conditions is an adaptation mechanism in response to the limited gas exchange that prevailed in these soil columns. These adventitious roots improve the diffusivity of gas along and across the root. In a study conducted at the Tasmanian Institute of Agriculture in Launceston, increased adventitious roots and aerenchyma formation were observed in cocksfoot and tall fescue under WL [33]. The decreased DM yield of ryegrass under the waterlogged moisture regime and subsequent effects on root growth were translated into decreased P offtake. The decreased DM yield and P offtake with increasing number of cuts is similar to long-term decrease of forage DM yield observed with other perennial grass species [34]. Increasing the number of cuts may increase the demand for plant growth resources such as light, water, and nutrients. With limited resources available, plants may allocate fewer resources to biomass production and nutrient uptake, resulting in decreased DM yield and P offtake. Despite high soil test P, as shown by Mehlich-3 P in the excess class, shoot DM yield response to slurry was significant. The increase in DM yield with increasing slurry could be explained by differences in the amount of other major and minor nutrients introduced by differences in manure rates, which result in an increase in P offtake. In a survey conducted across 85 farms with cropland, N input to the soil with manure application was on average 1364 kg N ha$^{-1}$ yr$^{-1}$, which is far above the mean annual N demands of the crops grown [35]. Manure application can result in accumulation of micronutrients such as copper and zinc in the soil top layer [36]. Manure application improves chemical and physical soil properties and stimulates biological activities, favoring biogenic aggregation and improving soil quality [37].

### 4.2. Effects of Soil Moisture Regimes, Slurry Rate, and Soil Depth on Reduction of Fe$^{3+}$ and Soil pH

The ORP in the topsoil was decreased under the waterlogged moisture regime due to the depletion of $O_2$ and $H^+$ concentrations. The decrease in ORP was also linked with changes in Fe forms as shown by the reduction of $Fe^{3+}$ to $Fe^{2+}$ [5]. Waterlogged soils experienced anaerobic conditions, which favor the partial decomposition of soil organic matter and the release of organic reducing substances. The extension of WL due to flooding can therefore affect chemical processes in the topsoil of agricultural lands. The consequences of decreased ORP due to flooding extend to changes in the chemistry or speciation of heavy metals in the soils. In our study, an emphasis was placed on the effects of changing ORP due to flooding on the reduction of $Fe^{3+}$ to $Fe^{2+}$. However, decreased ORP can also trigger the reduction of manganese (Mn) and sulfates with consequences for the environment. In contrast with the topsoil, ORP increased with increasing soil depth in soil columns under WL. This was not expected as increased ORP is an indication of an oxidative environment where the reduction of $Fe^{3+}$ into $Fe^{2+}$ and the consumption of $H^+$ are limited [5]. The Pearson correlation analysis highlighted negative relations between ORP and pH and $Fe^{2+}$ concentration, but positive relations between ORP and $Fe^{3+}$ concentration (Figure 10). It is possible that the volume of soil pores was limited at lower depths, thus restricting the excess water to saturate the soil matrix and induce reducing conditions. This assumption is supported by the observation of some unsaturated soil matrices during soil collection. Soil compaction is a common characteristic of soils in the Fraser Valley as a result of pedogenesis processes following the successive depositions of alluvial sediments, but also the use of heavy farm machinery during plowing and farm operations. Soil compaction is

enhanced by the use of heavy machinery when traffic occurs as the soil water content is above the plasticity limit. Soil compaction can cause impairments to the root system by increasing the soil bulk density, as well as water and gas flow in the soil by decreasing the porosity, especially the macroporosity. It is therefore possible that despite WL, the soil matrix was not saturated with water due to limited macroporosity, which maintained oxidative conditions and high concentrations of $Fe^{3+}$ at deeper soil layers. It is also possible that soil compaction restricts the access of microorganisms to alternative electron acceptors, such as $Fe^{3+}$ oxyhydroxides, which can lead to a decrease in the reduction process in the subsoil. There is also a potential for the presence of organic matter at the topsoil to stimulate the Fe reduction [38], thus decrease the ORP. This explanation is consistent with the negative relations between ORP and C and N, and the positive relations between soil pH and C and N (Figures 9 and 10). The role of organic substances is shown by decreased ORP, increased soil pH, and higher concentrations of $Fe^{2+}$ with increasing slurry rates. These changes could be linked to compounds such as organic matter and N present in dairy slurry (Figures 5c and 6d) (Table 3).

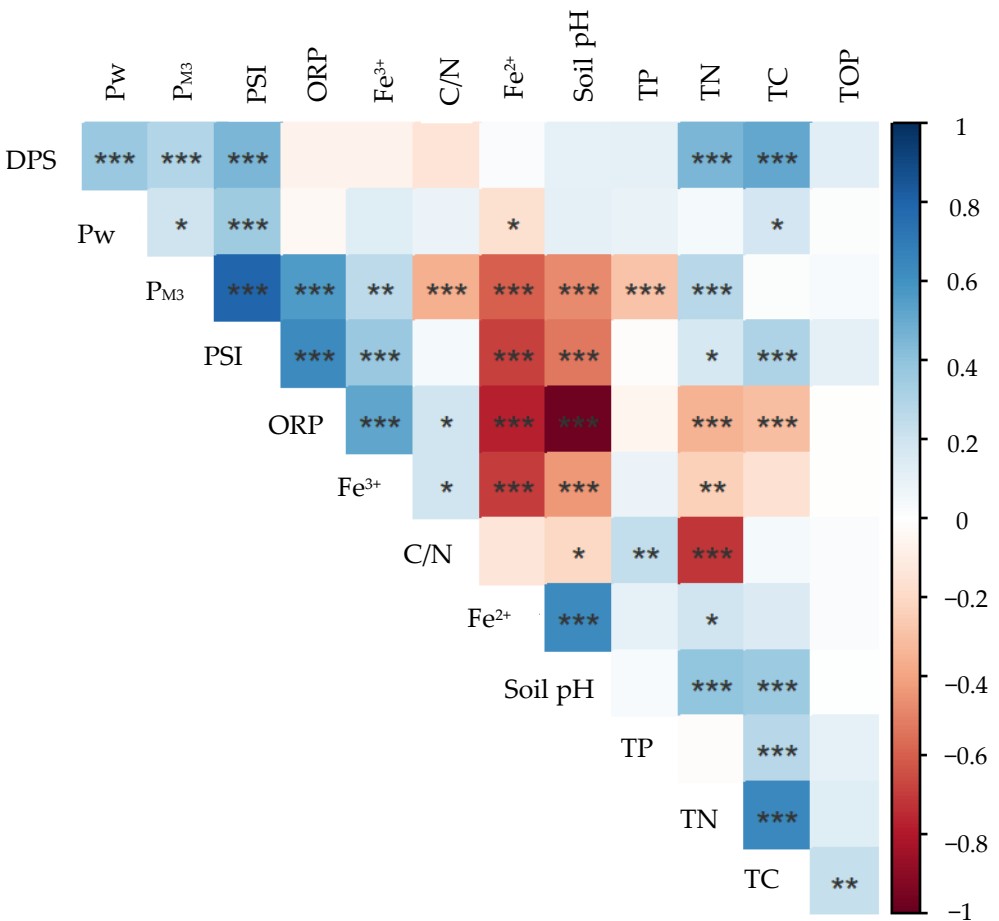

**Figure 10.** Correlation between the variables as determined by Pearson's correlation coefficient (Significant codes: '***' = $p < 0.001$, '**' = $0.001 < p < 0.01$, '*' = $0.01 < p < 0.05$). The colors and their intensity represent the correlation coefficient: blue indicates the positive correlation and red indicates the negative correlation. DPS = degree of phosphorus saturation; TOP = total organic phosphorus; TC, TN = total C and N; TP = total P; Soil pH = soil pH; $Fe^{2+}$ = ferrous concentration; C/N = C/N ratio; ORP = oxidation-reduction potential; $Fe^{3+}$ = ferric concentration; $P_{M3}$ = Mehlich-3 extractable P; PSI = phosphorus saturation index; Pw = water extractable P.

*4.3. Effects of Soil Moisture Regimes, Slurry Rate, and Soil Depth on Pw, PM3, and PSI*

The slight decrease in Pw in the topsoil (0–5 cm) under the waterlogged compared with field capacity moisture regime could be due to the dissolution of phosphate ions

in the excess water. During the 4-month period of WL, the concentration of Pw in the topsoil decreased by 30% compared with field capacity (Figure 7a). The topsoil, particularly the 0–5 cm, is the area of the soil where runoff water and phosphate ions interact. The extent to which phosphate ions were dissolved in the excess water could be attributed to the initial P status of the soil. The initial Pw concentration in the 0–15 cm soil depth was 9.4 mg kg$^{-1}$ (Table 1), which is higher than the critical threshold of Pw of 3.7 mg kg$^{-1}$ in the Fraser Valley [10]. Above the critical Pw, the risk of P loss with runoff water increases. Our results show that extended WL due to flooding could increase the risk of P loss in cropping systems with already high P status as found in the Fraser Valley due to a long history of over application of P fertilizer and manure. Our previous study has shown that P increases in floodwater with increasing WL events [5].

The trend of $P_{M3}$ with increasing moisture regime in the topsoil is paralleled with Pw (Figure 7b). The concentration of $P_{M3}$ in the topsoil decreased by 63% under waterlogged compared with field capacity. Similar observations could be made with PSI, where a decrease of 51% was obtained in the topsoil under waterlogged compared with field capacity (Figure 7e). The decrease in Pw, $P_{M3}$, and PSI in the topsoil under waterlogged compared with field capacity is consistent with the decreased ORP and $Fe^{3+}$, but increased $Fe^{2+}$ due to the reduction of $Fe^{3+}$ to $Fe^{2+}$ (Figures 5a and 6a,b). The reduction reaction of $Fe^{3+}$ to $Fe^{2+}$ leads to the release of phosphate ions fixed onto Fe oxides. The higher soil pH in the topsoil of soils under waterlogged compared with field capacity (Figure 4) is also consistent with the decreasing trend of soil P between the two moisture regimes (Figure 7a,c,e). The decrease in TOP under WL compared with FC suggests that the organic P was also dissolved from soils into excess water.

An important result of this study is the trend of Pw, $P_{M3}$ and PSI at lower soil depths. In soils under field capacity, there was a slight decrease in Pw, $P_{M3}$ and PSI between the topsoil (0–5 cm) and the lower soil layers (Figure 7a,c,e). In addition, Pw, $P_{M3}$, and PSI remained constant at depths 5–30 cm. The high Pw, $P_{M3}$, and PSI in the topsoil compared with the layers beneath in the soil under field capacity is consistent with the distribution of P in soils cropped with grass. In a study conducted across soils cropped with perennial grass in Canada, Finland, France, and Switzerland, soil test P concentrations were higher in the topsoil (0–5 cm) compared with lower soil depths [39]. The changes in soil P under field capacity can be attributed to the low mobility of P and the lack of mixing with residues and fertilizers. Greater soil P in the 0–5 cm soil layer than in deeper soil layers in soils cropped with grass were also reported in another study [40]. The soil used for this study was collected at a site that had not been under cultivation for decades. Our results indicate that stratification of P along the soil profile under a field capacity moisture regime may last for a long period. In a 37-year abandoned experiment on Nardus grassland in the Czech Republic, high concentrations of P were obtained in the topsoil compared with lower soil layers [41]. The persistence of P in the 0–5 cm soil layer can be attributed to variable surface charges present on the clay minerals on which phosphate ions adsorb easily and to soil organic matter liberating phosphate ions through decomposition and mineralization [40,41]. In contrast, in soils under a waterlogged moisture regime, Pw, $P_{M3}$, and PSI increased between the topsoil (0–5 cm) and the lower soil layers (5–30 cm) (Figure 7a,c,e). These trends indicate a downward movement of phosphate ions from the 0–5 cm layer to deeper soil layers (5–30 cm). It is possible that dissolved phosphate ions in the excess water under a waterlogged moisture regime were transported downward and accumulated at lower soil layers. In a first approximation, this explanation would not be consistent with the high ORP and overall limited reduction of $Fe^{3+}$ into $Fe^{2+}$ obtained in the deeper soil layers (Figures 5b and 6a,c), which indicates the existence of soil compaction that limits microbial activity involved in the reduction process. However, studies have shown that phosphate ions move downward along the soil profile through preferential flow in the unsaturated zone [7]. Crack flow is an important mechanism of preferential flow that occurs along continuous cracks in the soil. Phosphate ions can be transported downward through this pathway even though the soil matrix is unsaturated [7]. Phosphate ions that

accumulate at lower soil layers can be transferred into ditches and increase the risk of P loss following WL. One limitation of our study is the large gap of $P_{M3}$ in the topsoil between the two soil moisture regimes. There was a 63% decrease in $P_{M3}$ in the topsoil between the field capacity and waterlogged moisture regimes. We could not account for the decrease in $P_{M3}$ in the topsoil at lower soil layers (Figure 7c).

The concentrations of Pw and $P_{M3}$ and PSI had different behaviors following the application of dairy slurry. The increase in Pw under the field capacity regime with increasing dairy slurry rates is consistent with the literature [42]. In contrast, decreased $P_{M3}$ (Table 3) and PSI under WL (Figure 7f) with increasing dairy slurry rates could be attributed to the reduction of $Fe^{3+}$ to $Fe^{2+}$ triggered by increasing soil moisture and substances present in the slurry, as explained in the Section 4.2. This reduction process leads to the release of P and subsequently decreased $P_{M3}$ and PSI.

### 4.4. Effects of Soil Moisture Regimes, Slurry Rate, and P Offtake on P Loss

The concentration of phosphate ions in the leachate under FC did not exceed the critical limit to trigger freshwater algal blooms (0.05–0.10 mg P $L^{-1}$) [43] for dairy slurry application at 0, 15, and 30 kg P $ha^{-1}$. Although the leaching of phosphate did not show a significant difference across slurry rates, the average concentration of phosphate ions in the leachate with an application at 45 kg P $ha^{-1}$ under FC was 0.131 mg P $L^{-1}$, which may potentially trigger freshwater algal blooms. The initial $P_{M3}$ concentration in the topsoil was in the excess class (Table 1). However, the decreased Pw, $P_{M3}$, and PSI under FC at lower soil layers indicates a limited downward transfer of P (Figure 7a,c,e), which is consistent with the absence of a significant difference in phosphate ions in the leachate water across slurry rates. In addition, the total P offtake was on average 44.1 kg P $ha^{-1}$ under FC. This P offtake value indicates negative P balances for most dairy slurry applications except the highest application at 45 kg P $ha^{-1}$. The negative and close to zero P balances recorded across dairy slurry rates under FC explain the limited amount of P leached in our study. These results are consistent with the literature showing that P management strategies focusing on zero P balance are important to reduce the risk of P transfer from land to water sources [44].

In the case of WL, the decreased PM3 and PSI compared to the initial soil and FC indicate the movement of P from the soil surface into excess water. Moreover, the total P offtake was on average 31.2 kg P $ha^{-1}$ under WL, indicating negative and zero P balances for most dairy slurry applications except the highest application at 45 kg P $ha^{-1}$ that has a positive P balance. This result supports the hypothesis that P moved from the soil surface into excess water under WL, which could be exacerbated by a high application of slurry.

### 5. Conclusions

Excessive water content in the soil upon waterlogging decreased concentrations of $Fe^{3+}$, but increased concentrations of $Fe^{2+}$ and soil pH in the topsoil as a result of reduction reactions due to anaerobic conditions. These waterlogged soils also exhibited decreased concentrations of Pw, $P_{M3}$ and PSI in the topsoil and TOP across soil depth compared with field capacity. Therefore, the excess water dissolved phosphate ions and organic P that were fixed onto the soil matrix and organic matter. In contrast, there was no evidence of reducing conditions in deeper soil layers as shown by higher $Fe^{3+}$ concentrations compared with the topsoil. It is possible that the excess water in waterlogged soils did not saturate the soil matrix in deeper layers due to poor microporosity associated with soil compaction. This soil compaction may further hinder the access of microorganisms to alternative electron acceptors, including Fe, while the presence of organic matter in the topsoil could stimulate Fe reduction. Increased concentrations of Pw, $P_{M3}$, and PSI in deeper soil layers under a waterlogged moisture regime in the absence of reducing indicate that preferential flow played a key role in the downward transfer of P along the soil columns. These results confirm that a WL moisture regime associated with flooding changes the chemistry of metal cations such as $Fe^{3+}$ and enhances the risk of P loss from soil into water sources.

**Author Contributions:** Conceptualization, T.R. and A.J.M.; Methodology, T.R., A.J.M. and A.K.; Investigation, T.R.; Writing—original draft, T.R.; Writing—review and editing, T.R., A.J.M. and A.K.; Supervision, A.J.M. and A.K.; Funding acquisition, A.J.M. All authors have read and agreed to the published version of the manuscript.

**Funding:** Agriculture and Agri-Food Canada, A-Base project (ID: J-002266—Solutions for carryover of legacy P in the Fraser Valley and Hullcar Valley).

**Data Availability Statement:** The datasets generated for this study are available on request to the corresponding author.

**Acknowledgments:** We would like to acknowledge Agriculture and Agri-Food Canada for funding this project.

**Conflicts of Interest:** The authors declare no conflict of interest.

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
