# Peer review of "Phosphorus Mobility in Heavily Manured and Waterlogged Soil Cultivated with Ryegrass (Lolium multiflorum)"

_agronomy, doi:10.3390/agronomy13082168_

Round 1
Reviewer 1 Report
The study entitled: Phosphorus mobility in heavily manured and waterlogged soil cultivated with ryegrass (Lolium multiflorum) investigated how extended waterlogging in heavily manured soils affects soil phosphorus dynamics, ryegrass growth, and P leaching. Results showed reduced Fe3+ concentrations and increased Fe2+ and soil pH in the topsoil under waterlogging, potentially increasing P solubility and the risk of P loss with runoff and leaching. However, no evidence of reducing conditions was found in deeper soil layers. The excess water dissolved phosphate ions and organic P fixed onto the soil matrix, while preferential flow played a role in downward P transfer.
The manuscript is generally well-written, with a clear and coherent flow of idea except some typing errors as the duplication of brackets especially in material and method part.
Title is informative, and captures the essence of the study
The abstract provides a clear summary of the study, including the research question, methodology, key findings, and implications.
The introduction provides a clear and concise overview of the research problem and its significance, effectively setting the context for the study.
The methodology section is well-structured and sufficiently explains the research design, data collection, and analysis procedures in addition the authors supported the explanation with photos and illustrations.
Although the researchers dealt with the role of microbes and mentioned them in separate places in the research paper, they did not estimate the microbial activity during the stages of the experiment. I believe that measuring the microbial content in this research is important because of the difference in the microbial content between the different treatments from the ground up as a result of the different doses of dairy slurry between the treatments. but the paper can be published with the current status
The results are presented logically and in a visually appealing manner, making it easy to interpret the findings. And the tables and figures are clear and informative.
The discussion provides a comprehensive analysis of the results in light of the research question and relevant literature and supported by citation of recent references
The conclusion effectively summarizes the main findings and their significance in the broader context.
Author Response
Response to Comments: Reviewer #1
Question #1: The manuscript is generally well-written, with a clear and coherent flow of idea except some typing errors as the duplication of brackets especially in material and method part.
Response 1: The typing errors were addressed throughout the manuscript.
Question #2: Although the researchers dealt with the role of microbes and mentioned them in separate places in the research paper, they did not estimate the microbial activity during the stages of the experiment. I believe that measuring the microbial content in this research is important because of the difference in the microbial content between the different treatments from the ground up as a result of the different doses of dairy slurry between the treatments. but the paper can be published with the current status
Response 2: We totally agree that estimate of the microbial activity using MBC, MBN or MBP would have enhanced the quality of the manuscript. We will explore these parameters in future studies.
Reviewer 2 Report
Abstract
All abbreviations should be explained the first time they are used. For example - Pw, PSI.
Too much details, the main thing, tendencies, findings should be emphasized in the abstract. Much can be taken from the conclusions.
Introduction
References must be numbered in order of appearance in the text and listed at the end of the manuscript. This applies to the whole article in general.
The aim of the research is currently more like tasks - to determine something. You will definitely use this knowledge for something - to evaluate some risks, impact on the environment, etc. It is recommended to formulate the goal more broadly.
Materials and Methods
Table 1.
“Oxalate ammonium (FeOx, mmol kg–1)” – It seems to be a mistake, maybe it is oxalate Fe.
Results
“The DM yield increased with slurry rate (p = 0.013) from 17.4 g pot–1 at P0 to 21.7 g pot–1 at
P45 (Figure 3b).” - Which harvest time are we talking about? It is not clear.
Table 3. Are you sure there is no significant difference between PM3 for different moisture regimes? For both values there are “a”.
Figure 5c. Calculated redox potential - This part of the Figure 5 is empty, nothing is visible.
Figure 6. Figure 6b and Figure 6d – these parts are empty.
Figure 7. Also three parts – b, d and f are empty.
Apparently there is some technical error, pay attention and fix.
Discussion
Through the text, it is desirable to make completely sure that the conclusions, statements based on the results of your research and the findings of other authors are strictly separated, if only by the form of expression.
For example – first sentence “The decrease of shoot DM yield of ryegrass under waterlogged moisture regime can be attributed to physiological and metabolic changes of ryegrass (Kaur et al., 2020; Tian et al., 2021).” Is this a general finding, does it apply to your study or to the studies of the authors cited? In addition, the reference - Kaur et al., 2020 - is the first in the References section, although the other authors are listed in alphabetical order.
Conclusions successfully and concentrated summarize the information presented in the article. It would be valuable if the authors also focus on the possible directions of further research in this area, outline practical recommendations based on the acquired knowledge - how fertilizers should or should not be used in such wet soil conditions.
General conclusions
In general, the article is interesting, contains new knowledge and rich data material. Suggestions and recommendations for improving the quality of the article are provided in the comments to the authors. I recommend accepting this article in Agronomy after minor revision.
Author Response
Response to Comments: Reviewer #2
Question #1: Abstract
All abbreviations should be explained the first time they are used. For example - Pw, PSI.
Too much details, the main thing, tendencies, findings should be emphasized in the abstract. Much can be taken from the conclusions.
Response 1: We added the explanation of abbreviations that were used at the first time.
Question #2: Introduction
References must be numbered in order of appearance in the text and listed at the end of the manuscript. This applies to the whole article in general.
The aim of the research is currently more like tasks - to determine something. You will definitely use this knowledge for something - to evaluate some risks, impact on the environment, etc. It is recommended to formulate the goal more broadly.
Response 2: We applied numbered references in order of appearance in the text to the whole article and the list of the references was listed in order of appearance.
We also formulated the goal more broadly (in Abstract and introduction)
Question #3: Materials and Methods
Table 1. “Oxalate ammonium (FeOx, mmol kg–1)” – It seems to be a mistake, maybe it is oxalate Fe.
Response 3: We modified table 1. “Oxalate ammonium (Alox, mmol kg-1)” was replaced by “Oxalate ammonium aluminum (Alox, mmol kg-1)” and “Oxalate ammonium (Feox, mmol kg-1)” was replaced by “Oxalate ammonium iron (Feox, mmol kg-1)”.
Question #4: Results
“The DM yield increased with slurry rate (p = 0.013) from 17.4 g pot–1 at P0 to 21.7 g pot–1 at
P45 (Figure 3b).” - Which harvest time are we talking about? It is not clear.
Response 4: We added the explaination about which harvest time and under wich soil moisture regime we are talking about.
Question #5:
Table 3. Are you sure there is no significant difference between PM3 for different moisture regimes? For both values there are “a”.
Response 5: It was a mistake, there is significant different between PM3 for different moisture regimes. We modified the letter.
Question #6:
Figure 5c. Calculated redox potential - This part of the Figure 5 is empty, nothing is visible.
Figure 6. Figure 6b and Figure 6d – these parts are empty.
Figure 7. Also three parts – b, d and f are empty.
Apparently there is some technical error, pay attention and fix.
Response 6: We fixed the error for Fig 5, 6, and 7.
Question #7: Discussion
Through the text, it is desirable to make completely sure that the conclusions, statements based on the results of your research and the findings of other authors are strictly separated, if only by the form of expression.
For example – first sentence “The decrease of shoot DM yield of ryegrass under waterlogged moisture regime can be attributed to physiological and metabolic changes of ryegrass (Kaur et al., 2020; Tian et al., 2021).” Is this a general finding, does it apply to your study or to the studies of the authors cited? In addition, the reference - Kaur et al., 2020 - is the first in the References section, although the other authors are listed in alphabetical order.
Response 7: We modified the discussion following the comment.
Question #8: Conclusions successfully and concentrated summarize the information presented in the article. It would be valuable if the authors also focus on the possible directions of further research in this area, outline practical recommendations based on the acquired knowledge - how fertilizers should or should not be used in such wet soil conditions.
Response 8: We added the possible directions of further research and outline practical recommendations based on the acquired knowledge.